# Paranormal beliefs and cognitive function: A systematic review and assessment of study quality across four decades of research

**Charlotte E. Dean**[ID]*, **Shazia Akhtar, Tim M. Gale, Karen Irvine, Dominique Grohmann, Keith R. Laws**[ID]

Department of Psychology, School of Life and Medical Sciences, Sport and Geography, University of Hertfordshire, Hertfordshire, United Kingdom

* c.dean3@herts.ac.uk

**Data Availability Statement:** All data files relating to the quality assessment are available from the OSF repository (https://osf.io/7bthg/). Data relating

## Abstract

### Background

Research into paranormal beliefs and cognitive functioning has expanded considerably since the last review almost 30 years ago, prompting the need for a comprehensive review. The current systematic review aims to identify the reported associations between paranormal beliefs and cognitive functioning, and to assess study quality.

### Method

We searched four databases (Scopus, ScienceDirect, SpringerLink, and OpenGrey) from inception until May 2021. Inclusion criteria comprised papers published in English that contained original data assessing paranormal beliefs and cognitive function in healthy adult samples. Study quality and risk of bias was assessed using the Appraisal tool for Cross-Sectional Studies (AXIS) and results were synthesised through narrative review. The review adhered to the Preferred Reporting Items for Systematic Reviews and Meta-Analyses (PRISMA) guidelines and was preregistered as part of a larger registration on the Open Science Framework (https://osf.io/uzm5v).

### Results

From 475 identified studies, 71 (n = 20,993) met our inclusion criteria. Studies were subsequently divided into the following six categories: perceptual and cognitive biases (k = 19, n = 3,397), reasoning (k = 17, n = 9,661), intelligence, critical thinking, and academic ability (k = 12, n = 2,657), thinking style (k = 13, n = 4,100), executive function and memory (k = 6, n = 810), and other cognitive functions (k = 4, n = 368). Study quality was rated as good-to-strong for 75% of studies and appears to be improving across time. Nonetheless, we identified areas of methodological weakness including: the lack of preregistration, discussion of limitations, a-priori justification of sample size, assessment of nonrespondents, and the failure to adjust for multiple testing. Over 60% of studies have recruited undergraduates and 30% exclusively psychology undergraduates, which raises doubt about external validity.

to the 71 reviewed studies can be found within the paper's Supporting Information files.

**Funding:** The authors received no specific funding for this work.

**Competing interests:** The authors have declared that no competing interests exist.

Our narrative synthesis indicates high heterogeneity of study findings. The most consistent associations emerge for paranormal beliefs with increased intuitive thinking and confirmatory bias, and reduced conditional reasoning ability and perception of randomness.

## Conclusions

Although study quality is good, areas of methodological weakness exist. In addressing these methodological issues, we propose that authors engage with preregistration of data collection and analysis procedures. At a conceptual level, we argue poorer cognitive performance across seemingly disparate cognitive domains might reflect the influence of an overarching executive dysfunction.

## Introduction

The term "paranormal" typically refers to phenomena, such as psychokinesis, hauntings, and clairvoyance, which contradict the basic limiting principles of current scientific understanding [1]. Surveys consistently indicate paranormal beliefs are prevalent within the general population. For example, a representative survey of British adults conducted by the market-research company BMG Research [2] found that a third of their sample believed in paranormal phenomena, and a further 21% were 'unsure'. Of those who either believed in the paranormal or were unsure, 40% indicated they had seen or felt the presence of a supernatural entity. Similarly, Pechey and Halligan [3] found 30% of participants held at least one *strong* paranormal belief, and 79% held at least one paranormal belief at any strength (weak, moderate, or strong belief). Comparable levels of belief have been documented across various cultures over recent decades [4–7].

The most frequently used scales to measure paranormal beliefs include Tobacyk's Paranormal Belief Scale in both original (PBS) [8] and revised form (RPBS) [9], and the Australian Sheep-Goat Scale (ASGS) [10]. Despite widespread use, some concerns exist about both the content and the factor structures of these measures [11–13]. Nonetheless, both the RPBS and ASGS have demonstrated excellent internal reliability, with Cronbach's alpha values around .93 for the RPBS [14–16], and around .95 for the ASGS [17, 18].

Scores on paranormal belief measures have been linked to various personal and demographic characteristics. For example, higher belief scores have been noted for individuals high in extraversion and neuroticism [19–21], while lower belief scores have been seen for those with higher levels of education [22–24]. Paranormal belief levels also appear to vary across academic disciplines; with those engaged in hard (or natural) sciences, medicine, and psychology showing significantly lower paranormal belief scores than those in education, theology, or artistic disciplines [25, 26]. Higher levels of paranormal beliefs have been documented in women and younger individuals [27–32], though these sex and age effects are inconsistently reported [33] and have generated substantial debate [34–36].

### Paranormal beliefs and cognitive function

The association between cognitive functioning and paranormal beliefs has been researched over several decades. Such functions include memory, attention, language, and executive function (the umbrella term used to describe set-shifting ability, inhibitory control, and working memory updating; for a full description of executive function, see Miyake et al.'s work [37]).

As important for cognitive function is an individual's belief system. Religious and spiritual beliefs have been associated with slower cognitive decline in older adults [38, 39] but have also been shown to have an inverse relationship with memory performance [40] and intelligence [41, 42]. Similarly, so-called "epistemically unwarranted beliefs" [19], which includes belief in conspiracy theories, has been linked with lower educational attainment and reduced analytical thinking [43, 44]. Conspiracist beliefs are similarly associated with increased illusory pattern perception [45, 46], decreased need for cognition and cognitive reflection [47–49], biases against confirmatory and disconfirmatory evidence [50], and hindsight bias (for discussions on this topic see [51–53]).

The last published review to examine the relationships between paranormal beliefs and various aspects of cognition was conducted by Irwin in 1993 [53]. That non-systematic narrative review of 43 studies is now almost 30 years old and may have introduced bias by ". . .citing null results only when these form a substantial proportion of the available data on a given relationship" (p.6). At the time of his review, Irwin [53] concluded that, owing to the variable findings, support for the cognitive deficits hypothesis remained uncertain.

Research has grown considerably since Irwin's [53] review and an updated and systematic review is timely. The current review has two key aims: first, to provide the first assessment of study quality [54] in this area and second, to systematically review and summarise key associations between paranormal beliefs and a range of cognitive functions.

## Method

This review was conducted within the Preferred Reporting Items for Systematic Reviews and Meta-Analyses (PRISMA) guidelines [55] (see S2 Appendix for PRISMA checklist). The systematic review was preregistered at the Open Science Framework (OSF; https://osf.io/uzm5v) as part of a larger study (also assessing the relationships between paranormal beliefs and schizotypal personality traits). Data used for the descriptive and inferential analyses presented in the results section are available at the OSF preregistration. One author (CED) conducted the search strategy, article eligibility assessment, and data extraction.

### Search strategy

A systematic literature review was chosen for this area owing to its strength as a method to synthesise relevant evidence from large bodies of research [56, 57]. Our searches included both peer-reviewed articles published in scholarly journals and "grey literature" (concerning unpublished works such as doctoral theses).

We searched the electronic databases Scopus, ScienceDirect, SpringerLink, and OpenGrey from inception to May 2021. Our search terms were: (1) "paranormal belief" AND cogni*, (2) "paranormal belief" AND thinking, and (3) "paranormal belief" AND (memory OR "executive function"). For databases that did not permit wildcard Boolean operators (ScienceDirect), one of the above search terms was amended and entered as: "paranormal belief" AND (cognition OR cognitive), to best replicate the effect of the Boolean operator. Following exclusion of duplicate articles across databases, titles and abstracts were assessed to identify studies relevant to the review. Full-text assessment of eligible studies was performed to determine final inclusion. Full-text copies were unavailable for five studies, which were subsequently sought for retrieval. Finally, we hand-searched reference lists for each included article to identify any additional relevant articles. The PRISMA flow diagram presented in Fig 1 illustrates the full screening and selection process. The PRISMA checklist for abstracts is presented in S1 Appendix, and the full PRISMA checklist is presented in S2 Appendix.

**Fig 1. PRISMA flow diagram.**

### Inclusion/Exclusion criteria

Studies were eligible for inclusion if they were: published in the English language, conducted with a healthy adult sample (age 18 or over) and presented original data involving both a measure of paranormal belief and a measure of cognitive function. As cognitive functions have been shown to peak at different ages (for a detailed discussion on this topic, see [58]), we excluded samples that included children and adolescents under the age of 18 as some cognitive functions are still developing in these younger individuals.

### Data extraction

We used a detailed data extraction form to collate the following information from included studies: sample sizes and demographic details (including sex, age and education), the measures of self-rated paranormal belief, the aspect of cognition assessed, the tests of cognitive functions used, and findings relating to the relationship between paranormal beliefs and cognitive function. We categorised eligible outcome measures broadly to include both global cognitive function and domain-specific cognitive functions. Any measure of cognitive function was eligible

for inclusion (e.g., neuropsychological tests, self-report measures). Results for both paranormal beliefs and cognitive functioning could be reported as an overall test score that provides a composite measure, subscale scores that provide domain-specific measures, or a combination of the two. When multiple cognitive outcomes were investigated, we included all measures. To assess the strength of the relationships between paranormal beliefs and various cognitive functions, we calculated the number of positive, negative, or null findings reported by each study included in the review. Measures of paranormal belief were examined to determine the extent to which established questionnaires have been used.

In line with our preregistered protocol, we synthesised evidence narratively. Meta-analyses could not be undertaken because of the heterogeneity of study designs and outcome measures. We did, however, develop summary tables that include information relating to: sample size, gender composition, mean sample age, cognitive domain, outcome measure, and key findings. Given the range of outcome measures, we attempted to categorise the included studies by common cognitive domains. As the review took an explorative approach, and did not specify domains of interest, categorisation took place after full-text evaluation of included studies.

## Results

Electronic and hand searches identified 902 papers, of which 475 were unique. Most articles (k = 391) were excluded from the review following title and abstract screening, leaving 84 eligible for full-text evaluation. We removed 13 studies that included participants under the age of 18 (see S1 Table for details of these studies). Seventy-one papers met our inclusion criteria (see Fig 1), which included 70 published between 1980 and 2020 and one unpublished doctoral thesis [59].

### Assessment of study quality and risk of bias

The preregistration for this review specified using a bespoke series of questions to assess study quality, but we subsequently decided to use a more well-established and validated measure of study quality in the Appraisal tool for Cross-Sectional Studies (AXIS) tool [60]. Of the 20 AXIS items, seven assess reporting quality (items: 1, 4, 10, 11, 12, 16 and 18), seven relate to study design (items: 2, 3, 5, 8, 17, 19 and 20), and six to possible biases (items: 6, 7, 9, 13, 14 and 15). Two authors (DG and CED) independently rated each study, and these two sets of ratings had almost-perfect agreement (93%) with Kappa = .84.

Following previous research [61], we classified AXIS quality scores according to the number of "Yes" responses for the 20 items for each study—poor quality for scores <50%, fair quality for scores between 50 to 69%, good quality for scores of 70% to 79%, strong quality for scores of 80% and higher. Three in four studies were rated as either 'strong' (26/71: 37%) or 'good' (27/71: 39%). By contrast, 17/71 (24%) were rated as 'fair' and only 1/71 (1%) was rated as 'poor'. The mean quality rating score across all 71 studies was in the 'good' range; however individual AXIS items are not weighted and so this total score provides a general, but limited, classification that should be interpreted with some caution. The number of papers meeting each AXIS criterion ('Yes') is presented in Table 1. The number of papers meeting the criteria for each AXIS domain (reporting quality, study design quality, and potential biases) is presented in Figs 2–4 respectively.

All studies scored positively for items concerning: clear objectives, appropriate study design, appropriate measurement of outcome variables, internal consistency of presented results, and appropriate conclusions justified by the results. Study quality correlated with year of publication ($r$ = .64, $p < .001$), and appears to be improving with time (see Fig 5).

**Table 1. Total number of "yes", "no" and "unsure" responses for each AXIS item.**

| | AXIS Item | Yes | No | Unsure |
|---|---|---|---|---|
| **Introduction** | | | | |
| 1 | Were the aims/objectives of the study clear? | 71 | 0 | 0 |
| **Methods** | | | | |
| 2 | Was the study design appropriate for the stated aim(s)? | 71 | 0 | 0 |
| 3 | Was the sample size justified? | 5 | 66 | 0 |
| 4 | Was the target/reference population clearly defined? (Is it clear who the research was about?) | 68 | 3 | 0 |
| 5 | Was the sample frame taken from an appropriate population base so that it closely represented the target/reference population under investigation? | 22 | 49 | 0 |
| 6 | Was the selection process likely to select subjects/participants that were representative of the target/reference population under investigation? | 31 | 29 | 11 |
| 7 | Were measures undertaken to address and categorise non-responders? | 19 | 0 | 52 |
| 8 | Were the risk factor and outcome variables measured appropriate to the aims of the study? | 71 | 0 | 0 |
| 9 | Were the risk factor and outcome variables measured correctly using instruments/measurements that had been trialled, piloted or published previously? | 65 | 6 | 0 |
| 10 | Is it clear what was used to determine statistical significance and/or precision estimates? (e.g. p-values, confidence intervals) | 68 | 3 | 0 |
| 11 | Were the methods (including statistical methods) sufficiently described to enable them to be repeated? | 69 | 2 | 0 |
| **Results** | | | | |
| 12 | Were the basic data adequately described? | 66 | 5 | 0 |
| 13 | Does the response rate raise concerns about non-response bias? | 7 | 12 | 52 |
| 14 | If appropriate, was information about non-responders described? | 1 | 18 | 52 |
| 15 | Were the results internally consistent? | 71 | 0 | 0 |
| 16 | Were the results presented for all the analyses described in the methods? | 71 | 0 | 0 |
| **Discussion** | | | | |
| 17 | Were the authors' discussions and conclusions justified by the results? | 71 | 0 | 0 |
| 18 | Were the limitations of the study discussed? | 42 | 29 | 0 |
| **Other** | | | | |
| 19 | Were there any funding sources or conflicts of interest that may affect the authors' interpretation of the results? | 0 | 14 | 57 |
| 20 | Was ethical approval or consent of participants attained? | 37 | 0 | 34 |

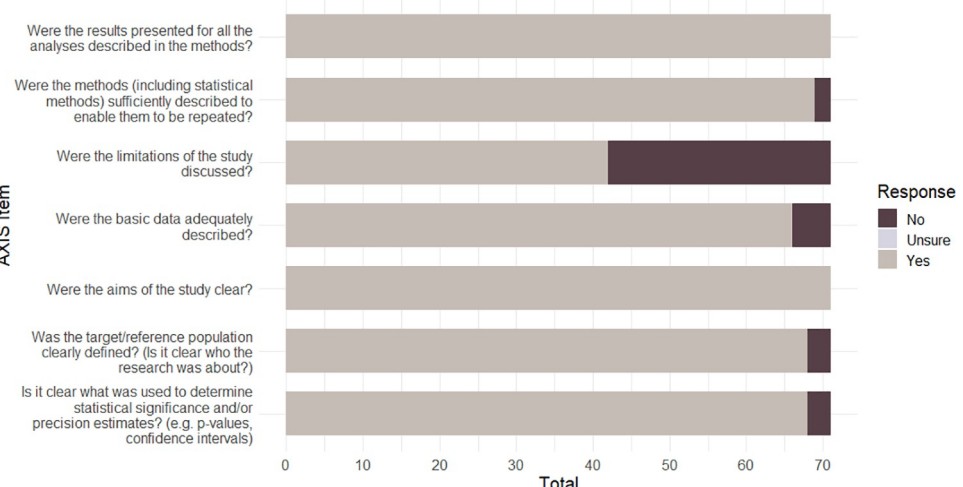

**Fig 2. AXIS reporting quality summary for the 71 papers included in the review.**

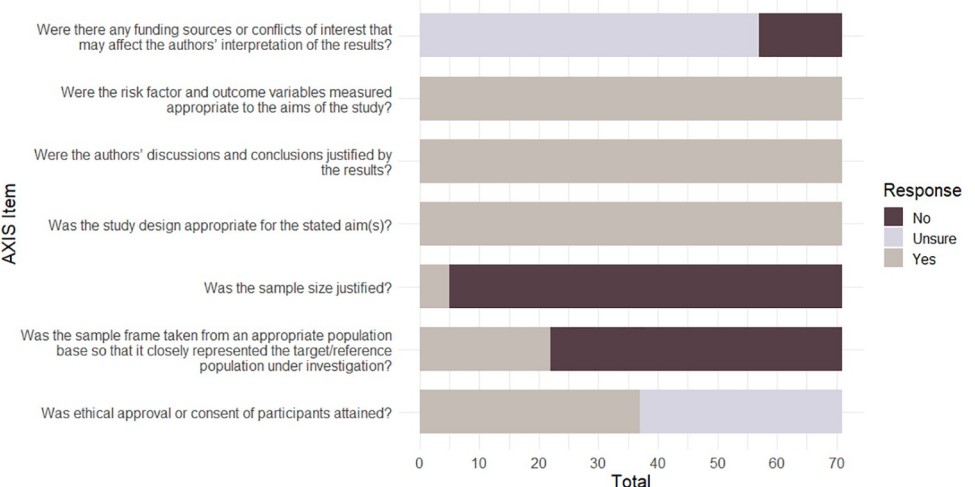

**Fig 3. AXIS study design quality summary for the 71 papers included in the review.**

Nonetheless, three main areas for study quality improvement were highlighted throughout the AXIS assessment: sample size justification, nonrespondents, and discussion of limitations.

## Sample size justification, sample representativeness and open science

Only 5 of 71 (7%) papers included a-priori power analyses to justify their sample sizes. Although power analyses are rarely conducted in this research area, the mean sample size is large at 211 (median = 124), suggesting that both simple correlational and between-subject comparisons are well-powered to detect large (.99 and .98), moderate (.94 and .88) and potentially for small effect sizes (.72 and .72)–large, moderate and small effects being 0.7, 0.5 and 0.2 respectively [62]. Despite this, many studies have assessed multiple outcomes and/or multiple metrics derived from the same tests and so, a simple power analysis will mislead. As a rough metric on this issue, we calculated the number of $p$-values presented in the results section for each of the 71 papers. This revealed a mean number of $p$-values per study of 43 (median = 30)

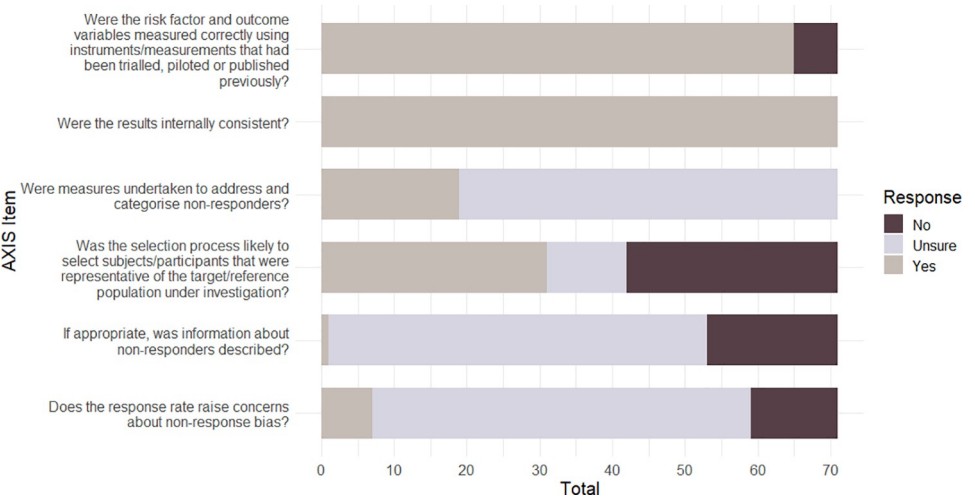

**Fig 4. AXIS possible biases summary for the 71 papers included in the review.**

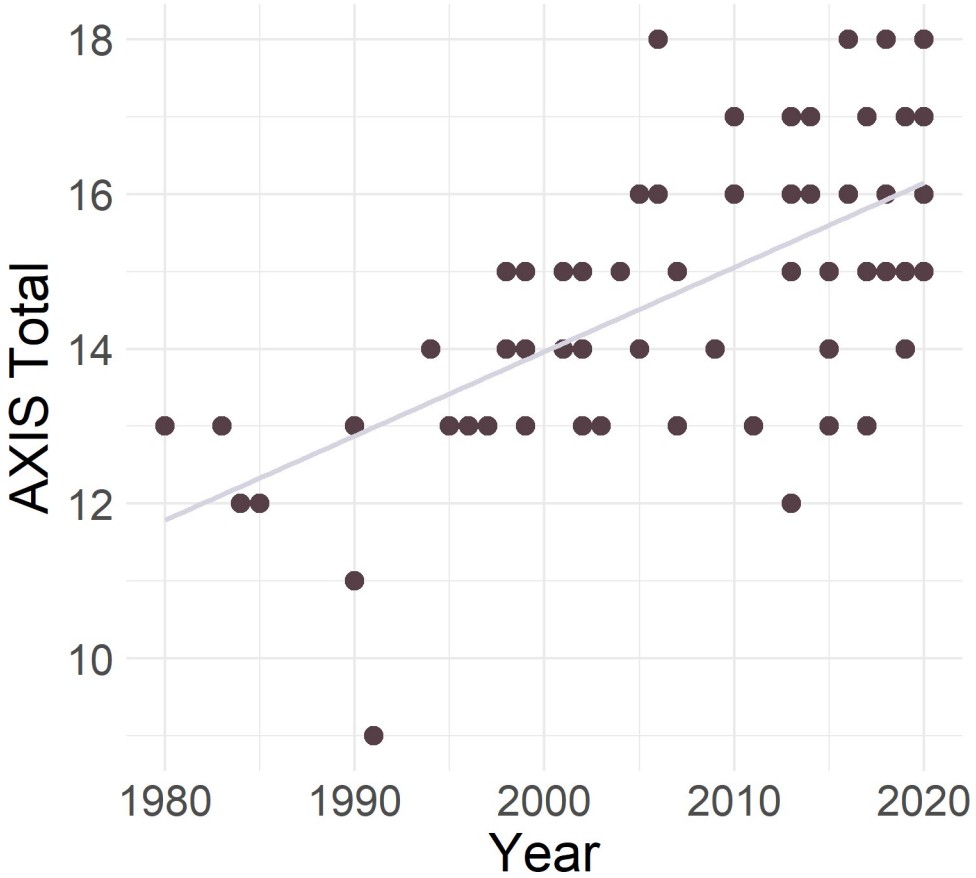

**Fig 5. AXIS study quality (maximum = 20) by year of publication.**

with a range from 1 [63] to over 200 [64]. So, despite relatively large samples, the possibility of type-1 errors remains high, especially when studies fail to adjust alpha levels for high levels of multiple testing. Only 12/71 studies employed some correction; eleven used a Bonferroni correction [15, 25, 64–72], and one used the Newman–Keuls adjustment [73]. Those studies that adjusted alpha levels tended to report more *p*-values than those that did not adjust (means 57 vs. 40). So, adjustment was made in fewer than one-in-five studies, most being published recently.

Despite good-strong quality ratings, some core features of open science practice including preregistration have yet to be embraced in this literature. Admittedly, we are assessing forty years of research and preregistration is a relatively recent innovation in psychology. Nonetheless, the Open Science Framework (OSF) began in 2013 as a repository for preregistrations–so potentially up to half of the 71 studies could have preregistered, yet only 2 (<3%) have done so [71, 74], with both published in 2020. The issue about preregistration is fundamental in this area of research. First, studies are characterised by large numbers of analyses often involving multiple outcome measures and/or multiple metrics derived from smaller numbers of tests. We have also seen that up to one-third of studies (25/71) have assessed relationships between cognitive function and paranormal test subscale scores (often with few items). This approach consciously or unconsciously increases the likelihood of reporting bias and HARKing (*hypothesizing after results are known*), often perhaps with little chance of, or interest in, replicating such findings (see Laws [75] for a discussion). Second, the preregistration of future trials will

also help to assess whether null results remain unpublished. Third, preregistration would identify both the primary outcome and the sample size required to achieve an acceptable level of statistical power. Ironically, the lack of attention to pre-registration and justifying sample sizes contrasts with research on paranormal phenomena, where study registration and a priori power calculations have been employed for many years [76].

**Representativeness.** Another issue concerns the sampling frame and its representativeness. Almost two-thirds of all samples are undergraduates (45/71: 63%) and of those, 21 (30%) consisted wholly of, or a majority of, psychology undergraduates. Only one-third of all samples consisted of: non-undergraduates (15/71: 21%), mixed undergraduate and general population samples (8/71: 11%) or other non-undergraduate samples (2/71: 3%). One non-undergraduate study by Blackmore in 1997 [77] consisted of a national newspaper-based study (Daily Telegraph) and recruited an exceptionally large sample (n = 6238). If we exclude this outlier, then 60% of all participants in the 70 remaining studies have been completely (k = 41) or majority undergraduate (k = 5) samples, with 16 involving only psychology graduates. Amongst the non-undergraduate samples, this includes visitors to a paranormal fair [29, 66], members of the Society for Psychical Research [78], Mechanical Turk participants [79], and some used Crowdflower, a crowdsourcing website [64, 80, 81]. So, even the non-undergraduate samples may not necessarily represent the wider population (see Stroebe et al. [82] for a discussion). Studies testing undergraduates and non-undergraduates did not differ in mean sample size (196 vs 215, with the exclusion of Blackmore [77], $t(68) = .29$, $p = .78$, $d = .08$) or in quality ratings (14.73 vs 15.19: $t(69) = -.90$, $p = .37$: $d = .23$). The profile of sampling, however, is pertinent because paranormal beliefs are inversely related to educational levels [22–24], and those studying sciences, medicine, and psychology exhibit lower levels of paranormal beliefs [25, 26]. Such samples are unrepresentative and may bias findings because they may combine lower levels of paranormal beliefs and higher cognitive functioning than occurs in the general population.

In addition to samples comprising more highly educated university students, most participants are female (>60%). The importance of this latter aspect of sampling is underscored for at least two reasons. First, some authors have documented greater levels of paranormal beliefs in women [27–32]. Indeed, the last literature review by Irwin in 1993 [53] stated that "the endorsement of most, but certainly not all, paranormal beliefs is stronger among women than among men" (p.8). Second, gender (and age) effects are not consistently reported [33] and have resulted in substantial debate [34–36]. This debate largely results from differences in psychological test theories (see Dean et al. 2021 [83] for a discussion). Classical test theory—used to develop common paranormal belief measures, such as the RPBS—does not test for the presence of differential item functioning (DIF). DIF refers to when individuals with the same latent ability (e.g., paranormal beliefs), but from different groups, have an unequal probability of giving a response. By contrast, modern test theory, including the use of Rasch scaling, can produce unbiased interval measures focused on the hierarchical properties of questionnaire items. This has resulted in the revision of older paranormal belief measures using modern test theory, to create scales that accurately capture fluctuations in levels of belief rather than differences in item functioning [84, 85]. When these problematic items are removed from scales such as the RPBS and ASGS, paranormal belief scores are no longer associated with sex, but small differences remain for age [84, 85]. Although these effect sizes seem to be small (e.g., 0.15 [84], identified by Cohen [62] as a small effect size), they are more likely to reflect a true and meaningful fluctuation in paranormal belief levels, compared to findings reported using scales developed through classical test theory.

**Nonrespondents.** Most studies (52/71) failed to state whether measures were undertaken to address and categorise nonrespondents. As such, response rates and risk of nonresponse

bias could not be calculated. Nonresponse bias arises when respondents differ from nonrespondents beyond sampling error and may reduce external validity [86, 87]. Survey-based approaches are at a greater risk of nonresponse bias owing to their high nonresponse rates, with those relying on self-administered online surveys suffering from higher nonresponse rates than those using face-to-face methods [88]. Most studies have been conducted in face-to-face settings (k = 59), however the past few years has seen a rise in online data capture (k = 12). Compared to face-to-face studies, online studies rated more highly on study quality (16.50 vs 14.49: $t(69)$ = -3.87, $p < .001$, $d = 1.32$) and had larger mean sample sizes (482 vs 155: $t(11.83)$ = -3.12, $p = .008$, $d = -1.69$, equal variances not assumed), but also report larger numbers of statistical comparisons (96.42 vs 31.58,: $t(12)$ = -3.47, $p = .005$, $d = 1.33$, equal variances not assumed).

Of the 19 papers that did provide nonresponse rates, seven had response rates < 70% and so raise concerns about potential nonresponse bias [89]. Only one of 19 papers [90] presented any information about nonrespondents, reporting that they had marginally lower educational attainment than respondents. Similar findings for nonrespondents have been reported in other research areas [91–94]. Finally, we note that online studies more often have records of nonrespondents. Guidance has been developed on detailing non-response details in online survey-type studies e.g., the Checklist for Reporting Results of Internet E-Surveys (CHER-RIES) [95] and should routinely be reported.

**Limitations.** Surprisingly, up to 40% of the included papers (29 of 71) did not include a discussion of study limitations. Discussion of study limitations forms a fundamental part of scientific discourse and is crucial for genuine scientific progress, allowing a reader to contextualise research findings [96]. The failure to discuss limitations might be viewed partly as a failure of the peer review process [97], but responsibility ultimately resides with authors. Detailing limitations allows other researchers to consider methodological improvements, identify gaps in the literature and has an ethical element by aiding research transparency. The inclusion of limitations not only helps increase research quality, but facilitates directions for future research and crucially, replications.

**Quality summary.** Of the 71 studies published since 1980, three-quarters were rated as 'good' or 'strong' in quality, and only one received a 'poor' quality rating. Indeed, study quality also indicates a continuous improvement in study quality across four decades of research. Despite the high levels of study quality and evidence of improving quality, we identified areas of methodological weakness: justifying sample size, providing more detail about non-respondents, and discussing study limitations.

One issue of note is the sampling, where almost two in three studies have relied on exclusively undergraduate samples (46/71: 65%), with many being psychology undergraduates. Future recruitment needs to move beyond the highly educated and address the bias towards female participants. Despite recruiting large samples, studies use large numbers of analyses, with a mean of 43 $p$-values reported in results sections, and rarely report appropriate adjustment of significance levels (12/71: 17%). These methodological issues are compounded by the fact that so few studies pre-register their primary hypotheses and analyses in advance (2/71: 3%).

## Cognitive functioning

The 71 studies were grouped into six sections: (1) perceptual and cognitive biases, (2) reasoning, (3) intelligence, critical thinking, and academic performance, (4) thinking style, (5) executive function, and (6) other cognitive functions. Whenever possible, categories were classified according to the focus identified by the authors in each study. Such classifications are

necessarily a simplification and not intended to provide a definitive organisation. Moreover, many studies could receive multiple classifications owing to the breadth of testing conducted (see S9 Table). In this context, S9 Table shows that two in three (48/71) studies might be classified as assessing executive function.

Articles presented in the first section (perceptual and cognitive biases) included scenarios aimed at measuring cognitive biases towards confirmatory evidence, and the impact of visually degraded stimuli on biases in perceptual decision-making. Examples of tasks used in the second section (reasoning) include the mental dice task [63] aimed at measuring probabilistic reasoning, and the Reasoning Tasks Questionnaire (RTQ) [98] to assess both probabilistic and conditional reasoning. Studies in the third category (intelligence, critical thinking, and academic performance) included published measures such as the Watson-Glaser Critical Thinking Appraisal (WGCTA) [99] and variations of Raven's matrices (e.g., the Advanced Progressive Matrices Test [100]; Raven's Progressive Matrices [101], and measures of academic achievement such as grade point average. In the fourth section (thinking style), papers used measures such as the Rational Experiential Inventory (REI) [102] and the Cognitive Reflection Test [103], aimed at assessing intuitive and analytical thinking. Studies in the fifth section (executive function and memory) included tasks such as the Deese-Roediger-McDermott task (DRM) [104] and the Wisconsin Card Sorting Test [105, 106]. The final cognitive section (other cognitive functions) included tasks to measure indirect semantic priming (using prime-target word pairs) and implicit sequence learning.

## Perceptual and cognitive biases

Nineteen articles (n = 3,397) assessed perceptual and cognitive biases. Perceptual decision-making with high visual noise stimuli has produced inconsistent findings (k = 7). For example, in 2014 Simmonds-Moore [67] found believers made more misidentifications of degraded black and white images of objects and animals (e.g., shark, umbrella), despite having faster response latencies than sceptics (suggesting a potential speed-error trade-off, with believers favouring speed over accuracy). By contrast, Van Elk [66] found sceptics mis-categorised degraded black and white images of face stimuli as houses more frequently than believers. The findings from both studies, however, contradict those from Blackmore and Moore's 1994 study [107], which reported no difference in the accurate identification of degraded monochrome images for believers and sceptics.

Two studies assessed perceptual decision-making relating to faces within degraded and artifact stimuli. Using black and grey images of faces and "nonfaces" (scrambled eyes-nose-mouth configurations), Krummenacher and colleagues [73] found believers made significantly more Type I errors than sceptics, favouring "false alarms" over "misses" (i.e., believers had a lower response criterion when classifying images as faces, with a bias towards "yes" responses). Similarly, Riekki et al. [108] presented participants with 98 artifact face pictures (containing a face-like area where eyes and a mouth could be perceived, e.g., a tree trunk) and 87 theme-matched non-face pictures (e.g., a tree trunk with no face-like areas). Believers rated the non-face pictures as more face-like and assigned more extreme positive and negative emotions to non-faces than sceptics.

A study conducted by Caputo [109] employed the strange-face illusion paradigm, in which pairs of participants are instructed to gaze into each other's eyes for 10 minutes in a dimly-lit room. This paradigm induces the experience of seeing face-related illusions and is assessed on a self-report measure (Strange Face Questionnaire; SFQ [110]). No association was found for paranormal beliefs and the experience of strange-face illusions. A final study of perceptual decision-making conducted by Van Elk [111] used point-light-walker displays (an animated-

point-set of 12 points, representing a human walking on a treadmill), randomly scrambling the location of each individual dot across the display; and participants had to detect if a human agent was present. Paranormal believers were more prone to illusory agency detection than sceptics, being biased towards 'yes' responses when no agent was present.

Cognitive biases have been assessed in 11 papers. These include reports of significant associations between paranormal belief and illusion of control or differences in causation judgements [65, 112–114] and risk perception [115]. Two studies, however, report no significant relationships [29, 116]. Further work shows that paranormal beliefs positively correlated with biases towards: anthropomorphism, dualism, teleology, and mentalising, but were not predicted by mentalising [15].

Proneness to jump to conclusions was assessed by Irwin and colleagues [68] using a computerised task [117]. Participants were informed of proportions of beads in two jars (e.g., 70 black and 30 red beads in jar one, but 30 black and 70 red beads in jar two), then shown a sequence of beads drawn one at a time from one of the jars and asked to identify whether beads were drawn from jar one or two, and to indicate when they are certain. Those who require fewer draws before being certain of their decision are identified as being prone to "jump to conclusions". A significant negative correlation emerged for jumping to conclusions, but only with the Traditional Religious Beliefs (TRB) subscale of the Rasch-devised RPBS [85]. A significant positive correlation was also found between TRB scores and self-report indices of jumping to conclusions as measured with the Cognitive Biases Questionnaire [118, 119] (e.g., "imagine you hear that a friend is having a party and you have not been invited", 1 = little or no inclination to jump to a premature conclusion, 2 = inclination to make a cautious inference, 3 = inclination to jump to a dramatic inference).

Prike et al. [64] assessed proneness to jumping to conclusions using both a neutral (beads task) and an emotional draws-to-decision task (where participants decide whether positive or negative words are more likely a description of "Person A" or "Person B"–for a full description see Dudley et al.'s work [120]). Participants also saw a series of 24 scenarios to assess bias towards confirmatory and disconfirmatory evidence, as well as liberal acceptance. Each scenario consisted of three statements presented one at a time, e.g., (a) "Eric often carries binoculars with him", (b) "Eric always has an unpredictable schedule", (c) "Eric tries to solve mysteries". Participants rated the likelihood of the same four response options after each statement, e.g., (a) "Eric is a private detective", (b) "Eric is a bird expert", (c) "Eric is a stalker", (d) "Eric is an astronaut". Each scenario presented an absurd interpretation (implausible for all three statements), a neutral lure, an emotional lure, and a true interpretation (less or equally as plausible as the lure options after the first statement but became the most plausible by the third statement). Paranormal beliefs were related to both disconfirmitory and confirmatory biases, but not to jumping-to-conclusions. Liberal acceptance predicted belief in the paranormal, but not after controlling for delusion proneness (as measured by the Peters et al. Delusions Inventory; PDI [121]). Lesaffre et al. [122] exposed participants to a magic performance and asked whether it was accomplished through: (1) paranormal, psychic, or supernatural powers, (2) ordinary magic trickery, or (3) religious miracles. Confirmation bias (i.e., explaining the magic performance in terms of paranormal powers) was associated with higher levels of paranormal beliefs. Barberia and colleagues [123] demonstrated that educating participants about confirmatory bias reduced scores on the Precognition subscale of the RPBS (but did not reduce global belief scores).

**Summary.** The studies assessing perceptual and cognitive biases are somewhat inconsistent regarding perceptual decision-making errors in response to degraded or ambiguous stimuli. Of the studies exploring perceptual decision-making, four suggest an inverse relationship between paranormal belief and perceptual decision-making, two found no

relationship, and one reported more perceptual decision-making errors from sceptics. Results show greater consistency when perceptual decision-making tasks involve identifying a human face/agent (rather than inanimate objects or animals), with believers making significantly more false-positive misidentifications than sceptics. In the 11 studies exploring cognitive biases, paranormal believers show a consistent bias towards both confirmatory and disconfirmatory evidence. The evidence that paranormal belief links to the tendency to "jump to conclusions" is weaker, but only two studies present findings related to this outcome.

## Reasoning

Seventeen papers have focussed on reasoning ability (n = 9,661), with the majority (12/17) reporting significant inverse relationships with paranormal beliefs and probabilistic reasoning. Perception of randomness and the conjunction fallacy have also been associated with paranormal beliefs on tasks with both neutral and paranormal content [69, 80, 124–128].

In 2007, Dagnall et al. [126] presented 17 reasoning problems across four categories: perception of randomness, base rate, conjunction fallacy, and probability. Perception of randomness problems required participants to determine the likelihood of obtaining particular strings (e.g., "Imagine a coin was tossed six times. Which pattern of results do you think is most likely? (a) HHHHHH, (b) HHHTTT, (c) HTHHTT, (d) all equally likely"). Performance on these problems significantly predicted paranormal belief, with believers making more errors than sceptics. No significant differences or predictive effects emerged for the three other problem categories. In a later study, Dagnall and colleagues [127] presented 20 reasoning problems across five categories of: perception of randomness, base rate, conjunction fallacy, paranormal conjunction fallacy, and probability. The authors again reported perception of randomness to be the sole predictor of paranormal beliefs, with high belief associated with fewer correct responses. While these papers report no effects in relation to conjunction fallacy, Rogers et al. [128] demonstrated a significant main effect of paranormal belief on conjunction errors, with believers making more errors than sceptics. In later studies, both Prike et al. [80] and Rogers et al. [129] reported an association between paranormal belief and conjunction fallacy, but this association was only significant for scenarios with confirmatory outcomes in the latter study.

Probabilistic reasoning ability has been consistently associated with paranormal beliefs across five studies. In one paper [130], participants received a probabilistic reasoning test battery comprised of six tasks. For example, one task was a variant of the birthday paradox (from Blackmore and Troscianko [97]), in which participants are asked: "How many people would you need to have at a party to have a 50:50 chance that two of them will have the same birthday (regardless of year of birth)". Possible answers for this task were 22 (correct), 43, or 98. Significant positive correlations emerged between paranormal beliefs and errors on three of the six tasks (dice sequences, dice throws, and sample size estimates). In the second study [63], participants received written descriptions of two hypothetical events: throwing 10 dice once to get 10 sixes and throwing one die 10 times to get 10 successive sixes; and had to identify whether one event was more probable or both equally probable. The authors reported 64% of believers and 80% of sceptics correctly identified that both events were equally probable. Brugger et al. [131] assessed differences in repetition avoidance between believers and sceptics on a mental dice task (where participants imagined throwing a die and had to write down the number they imagined being on top of the die), finding significantly fewer repetitions in believers than sceptics. Similarly, Bressan et al. [132] used a probabilistic reasoning questionnaire with problems concerning the comprehension of sampling issues, sensitivity to sample size, representative bias (as applied to sample size or random sequences) and the generation of random sequences. Believers made more probabilistic errors on two of four generation of random sequences

problems: (1) simulated coin toss problem, in which participants were asked to fill in 66 empty cells by writing 'H' (heads) or 'T' (tails) randomly to make a resulting sequence that was indistinguishable from that of an actually tossed coin), and (2) an adapted version of Brugger et al.'s [131] mental dice task. Finally, Blackmore [77] asked participants whether a list of 10 statements (as might be produced by a psychic, e.g., "there is someone called Jack in my family") were true for them, and to estimate the number of these statements that might be true for a stranger in the street. The number of 'true' statements was greater for believers than sceptics (significantly on five of the ten questions), however no significant differences emerged when estimating the number of statements true for a stranger.

The final four papers in this section found non-significant correlations between paranormal belief and *probabilistic* reasoning, but significant correlations with *conditional* reasoning tasks. Using the Reasoning Tasks Questionnaire (RTQ) [97], one study [4] found neither probabilistic reasoning nor *neutral* conditional reasoning were associated with paranormal beliefs. However, conditional reasoning was associated with paranormal beliefs when conditional reasoning tasks contained paranormal content rather than neutral content, with believers making fewer errors on these tasks. The second paper [133] measured reasoning using a test that combined probabilistic reasoning questions (seven in total, four of which were derived from the RTQ), conditional reasoning questions with abstract content (e.g., "if C is true, then D will be observed. D is observed. Therefore, C is true: True or False?"), and conditional reasoning questions with paranormal content (e.g., "if people are aware of hidden objects, then clairvoyance exists. People are aware of hidden objects. Therefore, clairvoyance does exist: True or False?"). Overall, paranormal beliefs correlated negatively with reasoning ability and conditional reasoning ability, but not with probabilistic reasoning ability. When comparing the two types of conditional reasoning questions, the authors reported no difference between the correlations for paranormal beliefs and either the abstract or paranormal conditions. Following a similar format, Wierzbicki [134] assessed reasoning ability using 16 conditional reasoning statements with either parapsychological or abstract content, finding paranormal belief scores and number of reasoning errors correlated positively. The final paper in this section [78] employed 32 statements conditional reasoning statements and found participants with strong paranormal beliefs made more reasoning errors than those with weak paranormal beliefs.

**Summary.** In general, evidence suggests paranormal beliefs are associated with poorer reasoning, however this line of research is characterised by inconsistent findings. Two studies report that the perception of randomness is a significant predictor of paranormal belief and provide some evidence of replicability [126, 127]. Despite this, evidence regarding the association between paranormal belief and the conjunction fallacy are conflicting, with two studies [127, 128] reporting no effect, and three [80, 128, 129] reporting significant associations. This may be due, in part, to the different statistical techniques used within each study, as those reporting no effect [126, 127] used multiple regression analyses with all probabilistic tasks entered as predictor variables, while studies reporting significant associations [80, 128, 129] only included conjunction fallacy tasks in their predictive models. Similar inconsistency emerges for probabilistic reasoning, with nearly equal numbers of studies reporting significant and nonsignificant associations with paranormal beliefs.

## Intelligence, critical thinking, and academic performance

Twelve studies explored intelligence, critical thinking, and academic performance (n = 2,657). Seven papers focused on critical thinking ability, with two finding significant reductions in paranormal belief following a course in critical thinking [70, 135]. Alcock and Otis' 1980 study

[136] employed the Watson-Glaser Critical Thinking Appraisal (WGCTA) [137] significantly higher levels of critical thinking ability in sceptics than believers. In 1998, Morgan and Morgan [138] conducted a similar study, measuring critical thinking using a revised version of the WGCTA [98], finding significant negative correlations between critical thinking ability and three subscales of the PBS (Superstition, Traditional Religious Belief, and Spiritualism). No significant correlation between paranormal belief and critical thinking emerged in the remaining three papers [139–141]. One did, however, report significant negative correlations between reasoning ability (measured using the Winer Matrizen-Test [142]) and three subscales of the PBS: Traditional Paranormal Beliefs, Traditional Religiosity, and Superstition [139].

The links between paranormal beliefs and academic achievement, or general intelligence are both mixed and weak. Two papers report significant negative correlations, one between overall paranormal belief scores and mean academic grade [25] and one between grade point average and the Witchcraft and Superstition subscales of the PBS [143]. Turning to intelligence, Betsch et al. [71] found a significant inverse relationship between IQ and paranormal beliefs, but only when controlling for sex, supporting similar findings from Smith et al.'s 1998 study [144] which reported a significant negative correlation between paranormal beliefs and intelligence (using the Advanced Progressive Matrices Test, Set 1 [100]). Nevertheless, two studies found no association between paranormal beliefs and intelligence. Royalty [141] used the information subtest of the Wechsler Adult Intelligence Scale [145] as an estimate of full-scale IQ, and the vocabulary subtest of the Multidimensional Aptitude Battery [146] as a measure of verbal intelligence. Stuart-Hamilton et al. [147] found no relationship with fluid intelligence using Raven's Progressive Matrices [101]; however, this sample were older (mean age of 71).

**Summary.**   Conflicting findings emerge from studies of intelligence, critical thinking, and academic performance, with an almost equal number of significant and non-significant associations to paranormal beliefs. Some of this heterogeneity, however, appears to reflect whether studies used crystallised or fluid intelligence tasks and the age of the sample (e.g., Stuart-Hamilton et al. [147] failed to find a relationship between fluid IQ and paranormal beliefs in an older sample, but Smith et al. [144] found a significant negative association in a younger sample). The precise relationship of paranormal belief with intelligence requires further investigation, both by considering the age of the sample and assessing relationships with fluid and crystallised intelligence separately.

## Thinking style

Thirteen studies (n = 4,100) examined aspects of thinking style. One consistent finding is a significant association between paranormal belief and an intuitive thinking style, which is characterised as being quick and guided by emotion [148–152]. A further study [153] also reports a significant partial correlation after controlling for sample type (online versus recruited face-to-face recruitment) owing to significantly higher levels of paranormal beliefs and intuitive thinking, and significantly lower rational/analytical thinking, in the online sample versus the face-to-face sample.

Contradictory findings, however, have emerged concerning paranormal beliefs and an analytical thinking style, which is thought to be more effortful and driven by logic. A positive relationship emerged in two studies [149, 150] while two [72, 152] found no relationship between paranormal beliefs and analytical thinking as assessed by the Rational Experiential Inventory (REI [102]). Four further studies report significant negative relationships between paranormal beliefs and analytical thinking using various measures: two [81, 154] used different versions of the Cognitive Reflection Test [103]; one [90] used the Rational Experiential Multimodal

Inventory [155]; and one [153] used both the Argument Evaluation Test [156] and the Actively Open-Minded Thinking scale [156, 157]. A further study reported a significant negative relationship between paranormal beliefs and analytical thinking but could not replicate the finding [74].

The final two papers in this section document relationships between paranormal belief and other cognitive styles. Gianotti et al. [158] presented participants with 80 word-pairs (40 semantically indirectly related, 40 semantically unrelated), and they had to state if a third noun was semantically related to both words. Believers showed increased verbal creativity, making significantly more rare associations than sceptics for unrelated word-pairs, but not for indirectly related word-pairs. Hergovich [159] used the Gestaltwahrnehmungstest [160] to assess degree of field dependence, by presenting participants with figures in which they needed to find an embedded figure in the form of a house and reported a significant positive relationship between paranormal beliefs and field dependence.

**Summary.**   Eight papers report positive associations between an intuitive thinking style and paranormal belief (although it should be noted that one study reported only a partial correlation after controlling for sample type). By contrast, evidence concerning an analytical thinking style is inconsistent, with reports of a negative relationship with belief (k = 4), a positive relationship (k = 2), and no relationship (k = 2). An additional study did report a negative relationship between analytical thinking and paranormal belief, but this was not replicated in a follow-up study. The final two studies in this section suggest positive relationships between paranormal belief and both verbal creativity and field dependence.

## Executive function and memory

Six studies (n = 810) assessed memory or executive function. Turning first to memory, the findings are inconsistent. One study [161] showed paranormal belief predicted false memory responses on a questionnaire-based measure, and two others [59, 78] reported associations between belief and behavioural measures of false memories but failed to replicate this in additional samples. Dudley's 1999 study [162] had participants complete the Paranormal Belief Scale while rehearsing a five-digit number or not; and found significantly higher paranormal belief scores in the group who had their working memory restricted (by the rehearsal task). However, a recent study by Gray and Gallo [79] failed to find any differences in working memory, episodic memory or autobiographical memory for believers and sceptics.

Further inconsistencies can be seen when exploring relationships between paranormal belief and inhibitory control, with Lindeman et al. [163] noting more errors from believers than sceptics on the Wisconsin Card Sorting Test [105, 106], but not on the Stroop task [164]. Wain and Spinella [165] explored executive function using a self-report measure and found a negative correlation between paranormal belief and executive functioning, with negative correlations between belief and both inhibition and organisation.

**Summary.**   The studies in this section report inconsistent links between paranormal belief and memory. While three of four memory studies report links between paranormal beliefs and an increased tendency to create false memories, two of these studies failed to replicate the finding. Two studies assessing executive functioning both suggest poorer performance is associated with belief but may interact with the measure of executive functioning.

## Other cognitive functions

Finally, four papers (n = 368) explored other aspects of cognitive function not covered by the categories already described. Pizzagalli et al. [166] tested the association between indirect semantic priming and paranormal beliefs using 240 prime-target word pairs, with target

words either directly related, indirectly related, or unrelated to the prime word. Compared to sceptics, believers had shorter reaction times for indirectly related target words were presented in the left visual field, suggesting a faster appreciation of distant semantic associations which the authors view as evidence of disordered thought. The final three papers did not find any significant relationships between paranormal beliefs and: implicit sequence learning [167], cognitive complexity [88], or central monitoring efficiency [168].

## General discussion

This systematic review provides the first evidence synthesis of the associations between paranormal beliefs and cognitive function since the early '90s [53] and the first assessment of study quality. The review identified 71 studies involving 20,993 participants. While most studies achieve good-strong quality ratings, specific areas of methodological weakness warrant further attention. In particular, studies often employ large numbers of measures, metrics and analyses, with no clearly identified primary outcome or adjustment of probability levels. These factors necessarily constrain any firm conclusions because of the high probability of Type 1 errors. Second, information about nonrespondents was either unreported or reported with insufficient detail to permit an assessment of potential nonresponse bias. Finally, up to a third of studies failed to discuss study limitations.

The cognitive deficits hypothesis is apparent in most papers (55/71), and a simple vote count shows that two-in-three studies (46/71) document that paranormal beliefs are associated with poorer cognitive performance. The most consistent findings across the six cognitive domains emerged between paranormal belief and an intuitive thinking style, with all eight studies confirming a positive association. Consistent findings also emerged for a bias towards confirmatory and disconfirmatory outcomes, as well as for poorer conditional reasoning ability and perception of randomness, though fewer studies were conducted in these areas. The two studies assessing executive functioning identified a negative association with paranormal belief but showed some inconsistency depending upon the type of executive test used. Associations with all other aspects of cognitive functioning (perceptual decision-making, jumping to conclusions and repetition avoidance, the conjunction fallacy, probabilistic reasoning, critical thinking ability, intelligence, analytical thinking style, and memory) have proven inconsistent, with nearly equal numbers of significant and null findings.

Various measurement issues, however, need to be considered. One concerns the large number of paranormal belief measures employed and their varied psychometric properties. The studies reviewed employed 26 different tests of paranormal belief, with the most common being the RPBS and a Rasch variant, with the next most common being 13 bespoke tests created by the authors. Such variability most likely contributes to heterogeneity across studies and potentially undermines the reliability of reported associations between cognitive functions and paranormal beliefs. For a full summary of the scales used in each study, see S8 Table.

Not only does the range of *cognitive* measures used within each cognitive domain contribute to heterogeneity across studies, but so does the reliability of such measures. As Hedge et al. [169] note, individual differences in relation to cognition and brain function often employ cognitive tasks that have been well-established in experimental research. Such tasks may not be directly adaptable to *correlational* research, however, for the very reason that they elicit robust experimental effects; they are specifically designed and selected for low between-participant variability. Most studies presented here are correlational and use a combination of established experimental tasks (e.g., the WCST, Raven's Matrices, Cognitive Reflection Test, Embedded Figures Test) and questionnaire-based methods to assess cognition. This may undermine the reliability of reported associations between cognitive functions and paranormal

beliefs if studies use experimentally derived cognitive tasks that are sub-optimal for correlational studies. Hedge et al. [169] offer several suggestions to overcome this, such as the use of alternative statistical techniques (e.g., structural equation modelling), factoring reliability into a-priori power calculations to reduce the risk of bias towards a null effect, or using within-subjects designs when the primary goal of the study is to examine associations between measures rather than focusing on individual differences per se. The largely correlational approach of studies reviewed here also suffers from the standard limitations of questionnaire studies and correlational designs. Although regression approaches can be powerful, they cannot establish causality without the use of longitudinal methods. This correlational approach also means that moderators and mediators of the relationship between paranormal beliefs and cognition remain underspecified.

## Future directions–the fluid-executive model

The general trend of the current review accords with the cognitive deficits hypothesis approach described by Irwin almost 30 years ago [53]–at least insofar as around 60% of published studies document paranormal beliefs to be associated with poorer cognitive performance. Nonetheless, the cognitive deficits hypothesis does not provide an entirely satisfying account of why paranormal believers and sceptics perform differently on such a wide variety of cognitive tasks. This has some key implications: first, that people who believe in the paranormal seemingly have a disparate array of cognitive deficits–are these assumed to have occurred independently of each other or do they somehow accumulate various cognitive deficits? Another implication is that such an array of cognitive deficits is largely atheoretical, with various researchers pursuing seemingly independent lines of research linking cognitive function to paranormal beliefs with little attention to integration. Hence a somewhat underspecified model pervades the literature, with often limited justification for the specific role played by cognitive function in paranormal beliefs or how and why such an array of deficits are identifiable in paranormal believers. Given the almost complete lack of preregistration, accompanied by the large numbers of statistical analyses often conducted without correction, we also cannot exclude concerns about potential publication bias, false positives, and selection bias. Empirical studies presenting significant or favourable findings are, of course, more likely to be published [170]; and crucially, psychologists tend to rate studies as having better quality when they conform to prior expectations. Hergovich et al. [171] demonstrated this bias by presenting psychologists (all of whom did not believe in astrology) with descriptions of parapsychological studies, finding that they gave higher quality ratings to studies disproving astrological hypotheses. Participants were less likely to complete the study if they received an abstract confirming astrological hypotheses, with an attrition rate of 38.90%. These issues underscore the importance of pre-registered replications of key findings (see Laws [172] for a discussion). To our knowledge, potential publication bias has not been extensively assessed. A previous meta-analysis of psychokinesis studies indicated the presence of publication bias [173], but this claim has been challenged [174]. Finally, questions also arise about whether poorer performance by believers on any cognitive ability tests even merits the descriptor of 'deficits'; and recently has been rephrased more neutrally as the cognitive *differences* hypothesis [79]. The term 'deficit' typically implies a *permanent* lack or loss of cognitive function; however, little to no research has looked at the consistency of cognitive performance in paranormal believers across time and established whether poorer cognitive performance is more trait than state dependent. While paranormal beliefs appear to be largely trait-like, they may have a state component [175].

While current studies do not necessarily endorse Irwin's 1993 [53] comment that "...the believer in the paranormal is held variously to be illogical, irrational, credulous, uncritical, and foolish" (p.16), they converge on an underlying non-specific cognitive deficit or collection of deficits. Typically, when an array of cognitive deficits/differences are documented, researchers would want to know if specific areas of cognitive weakness emerge. Currently, no cognitive area suggests a specific deficit profile in paranormal believers. Although not directly tested, paranormal believers might display heterogeneous cognitive profiles that link to different paranormal belief components. Nonetheless, it is hard to see why or how specific types of paranormal belief content would link to different cognitive deficits.

One possibility is that the failure of any specific area of cognitive dysfunction to emerge (amongst perceptual and cognitive biases, reasoning, intelligence, critical thinking and academic performance, thinking style, and executive functioning), may point to a common shared underlying cognitive component. One feasible interpretation is that many of the tasks described in the various domains described here do in fact share a common cognitive ability—higher-order executive functions (planning, reasoning and problem-solving, impulse control, initiation, abstract reasoning, and mental flexibility), which in turn may be related to aspects of fluid intelligence [176].

Human functional brain imagining identifies strikingly similar patterns of prefrontal cortex activity in response to cognitive challenges across various seemingly different domains, including: increased perceptual difficulty (high vs low noise degradation), novelty, response conflict, working memory, episodic and semantic memory, problem solving, and task novelty [177–179]. This demand-general activity underlies our ability to engage in flexible thought and problem-solving [177] and is closely linked to fluid intelligence [180]. We propose that the broad cognitive-deficit profile linked to paranormal beliefs may overlap with functions of the multiple-demand (MD) system. Part of the function of the MD system concerns its role in the separation and assembly of task components and that this accounts for the link with fluid intelligence. In this context, we suggest that each of the cognitive domains linked to paranormal beliefs may indeed be subserved by this MD system housed in the fronto-parietal cortex. The section on executive function is self-evidently linked with the frontal system. The section on intelligence similarly highlights links between paranormal beliefs and fluid IQ measures such as the Ravens Matrices [100, 101]. Studies further show the same MD system is recruited when confronted with perceptually difficult tasks (such as those outlined in the section on perceptual and cognitive biases for degraded visual input) [66, 67, 107, 108]. Aside from supporting our problem-solving ability, fluid intelligence and various aspects of executive functioning (e.g., working memory) underpins our ability to reason and to see relations among items and includes both inductive and deductive logical reasoning. The section on reasoning shows paranormal beliefs are related to conditional and probabilistic reasoning [69, 77, 80, 124–134]. Thus, many of the cognitive deficit-paranormal belief associations may be reframed as the product of a single underlying fluid intelligence-executive component. Going forward, such a model suggests potential avenues of research. One prediction would be that groups of believers and sceptics matched for fluid IQ would be less likely differ on a range of cognitive tasks.

## Limitations of the present review

The current review is the first to assess the quality of studies examining cognitive function and paranormal beliefs. We report study quality is good-to-strong, with interrater reliability on AXIS ratings being almost-perfect (93%). Individual AXIS items however are not weighted and any simple comparisons between specific studies across total summed quality scores should be regarded with caution [181–183]. Thus, two studies with the same total quality

score, but across different items, might not be comparable because some items may be more concerning to quality than others. Hence, we have focused on specific domains of strength or weakness across studies.

We acknowledge substantial limitations regarding the classification of studies into six areas of cognitive function: (1) perceptual and cognitive biases, (2) reasoning, (3) intelligence, critical thinking, and academic performance, (4) thinking style, (5) executive function, and (6) other cognitive functions. S9 Table shows that many of the studies could be re-classified and indeed, two-thirds (48/71) could be re-classified as assessing executive functioning. The latter is consistent with our proposal that a substantial proportion of the published studies may be documenting a relationship between paranormal beliefs and higher-level executive function/ fluid intelligence.

Our preregistered protocol had an exclusion criterion concerning samples with individuals aged less than 18, and this led to our excluding 11 datasets (see S1 Table for a complete list and details; Aarnio & Lindeman [26], Saher & Lindeman [184], and Lindeman & Aarnio [185] were overlapping or identical samples). A key reason for exclusion was because age impacts both cognitive functions and paranormal beliefs. Certain cognitive functions, for example executive functions, take until late adolescence or early adulthood to mature [186]. Additionally, younger individuals also show higher levels of paranormal beliefs [187; for a discussion see Irwin's review, 53]. While the exclusion of these studies is a potential limitation, their exclusion does not change our key findings or conclusions drawn from this review. In the same context, our lack of an upper age limit exclusion criterion could also be considered as a limitation. Sixteen papers (23%) reviewed here included participants aged 65+ (though 25/71 (36%) studies did not report on the age range of participants). While some cognitive functions do not mature until late adolescence or early adulthood, measurable changes in cognitive function occur with normal aging. Performance on certain cognitive tasks has been shown to decline with age, such as those requiring executive functioning (including decision-making, working memory and inhibitory control), visuoperceptual judgement and fluid-intelligence [188, 189]. Such cognitive declines have been associated with age-related reductions of white matter connections in brain regions including the prefrontal cortex [190, 191].

Finally, one limitation is that we were unable to conduct a meta-analysis because of the large variability in outcome measures within and between studies, which make it challenging to determine the precise outcome being tested. In parallel, the large numbers of analyses per study also mean that conclusions from our systematic review regarding findings for specific cognitive domains must also be interpreted with some caution.

## Conclusions

Our systematic review identified 71 studies spanning: perceptual and cognitive biases, reasoning, intelligence, critical thinking, and academic performance, thinking styles, and executive function. However, then tasks employed to assess performance in each domain often appear to require higher-order executive functions and fluid intelligence. We therefore propose a new, more parsimonious, fluid-executive theory account for future research to consider. Methodological quality is generally good; however, we highlight specific theoretical and methodological weaknesses within the research area. In particular, we recommend future studies preregister their study design and proposed analyses prior to data collection, and address both the heterogeneity issues linked to paranormal belief measures and the reliability of cognitive tasks. We hope these methodological recommendations alongside the fluid-executive theory will help to further progress our understanding of the relationship between paranormal beliefs and cognitive function.

## Supporting information

**S1 Appendix. PRISMA abstract checklist.**
(DOCX)

**S2 Appendix. PRISMA checklist.**
(DOCX)

**S1 Table. Papers excluded from the review (participants < 18).** Note: Ts = Thinking Style, CPb = Cognitive and Perceptual Biases, O = Other Cognitive Functions, REI = Rational and Experiential Inventory (Epstein et al., 1996), SJQ = Scenario Judgements Questionnaire (Rogers et al., 2016; Rogers et al., 2011), IPO-RT = Inventory of Personality Organization (Lenzenweger et al., 2001), RT = reality testing, ASGS = Australian Sheep-Goat Scale (Thalbourne & Delin, 1993), ESP = extrasensory perception, LAD = life after death, PK = psychokinesis, NAP = new age philosophy, TPB = traditional paranormal beliefs, RPBS = Revised Paranormal Belief Scale (Tobacyk, 2004; Lange et al., 2000), CKCS = Core Knowledge Confusions scale (Lindeman & Aarnio, 2007; Lindeman et al., 2008), CRT = Cognitive Reflection Test (Frederick, 2005), BRC = base-rate conflict, BRN = base-rate neutral, SREIT = Self-Report Emotional Intelligence Test (Schutte et al., 1998), WCQ = Ways of Coping Questionnaire (Folkman & Lazarus, 1988), IBI = Irrational Beliefs Inventory (Koopmans et al., 1994).
(DOCX)

**S2 Table. Studies included in the systematic review concerning perceptual and cognitive biases.** Note: / = information not reported, P = perceptual biases, C = cognitive biases, bl = believers, sc = sceptics, + = positive,— = negative, corr. = correlation, Ns. = nonsignificant, ESP = extrasensory perception, BADE = bias against disconfirmatory evidence, BACE = bias against confirmatory evidence, TRB = traditional religious beliefs, ELF = extraordinary life-forms, PRI = Personal Risk Inventory (Hockey et al., 2000), SFQ = Strange-Face Questionnaire (Caputo, 2015), IDAQ = Individual Differences in Anthropomorphism Quotient (Waytz et al., 2010), DS = Dualism Scale (Stanovich, 1989), EQ = Empathy Quotient (Baron-Cohen & Wheelwright, 2004).
(DOCX)

**S3 Table. Studies included in the systematic review concerning reasoning.** Note: / = information not reported, + = positive,— = negative, corr. = correlation, Ns. = nonsignificant, ESP = extrasensory perception, PK = psychokinesis, LAD = life after death, NAP = new age philosophy, DR = deductive reasoning, RTQ = Reasoning Task Questionnaire (Blackmore & Troscianko, 1985), ASGS = Australian Sheep-Goat Scale (Thalbourne & Delin, 1993), RPBS = Revised Paranormal Belief Scale (Tobacyk, 2004), MMU-N = Manchester Metropolitan University New (Dagnall et al., 2010).
(DOCX)

**S4 Table. Studies included in the systematic review concerning intelligence, critical thinking, and academic performance.** Note: / = information not reported, C = cognitive ability, I = intelligence, m = males, f = females, + = positive,— = negative, corr. = correlation, Ns. = nonsignificant, ATS = Assessment of Thinking Skills (Wesp & Montgomery, 1998), WGCTA-S = Watson-Glaser Critical Thinking Appraisal Form S (Watson & Glaser, 1994), WGCTA = Watson-Glaser Critical Thinking Appraisal (Watson & Glaser, 2002; Watson & Glaser, 1980; Watson & Glaser, 1964), RPM = Raven's Progressive Matrices (Raven et al., 2000), RPM Rasch Model = Raven's Progressive Matrices Rasch Model (Rasch, 1960), MHVT = Mill Hill Vocabulary Test (Raven et al., 1998), CCTT = Cornell Critical Thinking Test (Ennis & Millman, 1985), WMT = Wiener Matrizen Test (Formann & Piswanger, 1979),

APM = Advanced Progressive Matrices (Raven, 1976), WAIS-IS = Wechsler Adult Intelligence Scale Information Subtest (Wechsler, 1955), GPA = Grade Point Average.
(DOCX)

**S5 Table. Studies included in the systematic review concerning thinking style.** Note: / = information not reported, + = positive,— = negative, corr. = correlation, Ns. = nonsignificant, AOT = Actively Open-Minded Thinking Scale (Stanovich et al., 2016; Stanovich, 1999), CRT = Cognitive Reflection Test (Frederick, 2005), CRT-2 = Cognitive Reflection Test-2 (Thompson & Oppenheimer, 2016), REI = Rational-Experiential Inventory (Pacini & Epstein, 1999), WST = WordSum Test (Huang & Hauser, 1998), RI = Rational/Experiential Inventory (Norris & Epstein, 2011), IPSI-SF = Information-Processing Style Inventory Short Form (Naito et al., 2004), FIS = Faith in Intuition Scale (Pacini & Epstein 1999), NFC = Need for Cognition scale (Cacioppo et al., 1984), AET = Argument Evaluation Test (Stanovich & West, 1997), 10-Item REI = 10-Item Rational-Experiential Inventory (Epstein et al., 1996), GWT = Gestaltwahrnehmungs Test (Hergovich & Hörndler, 1994), EFT = Embedded Figures Test (Witkin et al., 1971).
(DOCX)

**S6 Table. Studies included in the systematic review concerning executive function and memory.** Note: / = information not reported, M = memory, EF = executive function, bl = believers, sc = sceptics, + = positive,— = negative, corr. = correlation, Ns. = nonsignificant, DRM = Deese-Roediger-McDermott (Roediger & McDermott, 1995), CRT = Criterial Recollection Task (Gallo, 2013), IIT = Imagination Inflation Task (Garry et al., 1996), RSPAN = Reading-Span Task (Daneman & Carpenter, 1980), OSPAN = Operation Span Task (Turner & Engle, 1989), SILS = Shipley Institute of Living Scale (Zachary, 1986), AET = Argument Evaluation Task (Stanovich & West, 1997), RAT = Remote Associations Test (Mednick, 1962), WCST = Wisconsin Card Sorting Test (Berg, 1948; Grant & Berg, 1948), EFI = Executive Function Index (Spinella, 2005), ANP = anomalous natural phenomena, TRB = traditional religious beliefs, NCQ = News Coverage Questionnaire (Wilson & French, 2006), ASGS = Australian Sheep-Goat Scale (Thalbourne 1995; Thalbourne & Delin, 1993), AEI = Anomalous Experiences Inventory (Kumar et al., 1994).
(DOCX)

**S7 Table. Studies included in the review concerning other cognitive functions.** Note: / = information not reported, bl = believers, sc = sceptics, f = females, m = males, ISL = implicit sequence learning, ISP = implicit semantic priming, VF = visual field, LVF = left visual field, RVF = right visual field, CME = central monitoring efficiency, RE = reasoning errors, CC = cognitive complexity, + = positive,— = negative, corr. = correlation, Ns. = nonsignificant, SPQ-B = Schizotypal Personality Questionnaire Brief (Raine & Benishay, 1995), RCRG = Role Construct Repertory Grid (Kelly, 1955).
(DOCX)

**S8 Table. Measures of paranormal beliefs used in the 71 studies included in the review.** Note:† = papers that provided reliability statistics for their novel scales, ‡ = used a translated version of the original scale, * = Musch & Ehrenberg (2002) developed a novel scale that was later named the BPS and was used in two subsequent studies. RPBS = Revised Paranormal Belief Scale (Tobacyk 1988; 2004), ASGS = Australian Sheep-Goat Scale (Thalbourne & Delin, 1993), PBS = Paranormal Belief Scale (Tobacyk & Milford, 1982), Rasch RPBS = Rasch devised Revised Paranormal Belief Scale (Lange et al., 2000), BPS-O = Belief in the Paranormal Scale (Original; Jones et al., 1977), BPS = Belief in the Paranormal Scale (Musch & Ehrenberg, 2002), MMU-N = Manchester Metropolitan University New (see Dagnall et al., 2010),

MMU-PS = Manchester Metropolitan University Paranormal Scale (see Dagnall et al., 2010),
SSUB = Survey of Scientifically Unsubstantiated Beliefs (Irwin & Marks, 2013),
OS = Occultism Scale (Böttinger, 1976), PS = Paranormal Scale (Orenstein, 2002),
AEI = Anomalous Experiences Inventory (Gallagher et al., 1994; includes a 'belief' subscale).
(DOCX)

**S9 Table. Alternate categorisations for studies included in the review.** Note: ✓ = original category, ✓ = alternate category.
(DOCX)

## Author Contributions

**Conceptualization:** Charlotte E. Dean.

**Data curation:** Charlotte E. Dean.

**Formal analysis:** Charlotte E. Dean, Dominique Grohmann, Keith R. Laws.

**Investigation:** Charlotte E. Dean.

**Supervision:** Shazia Akhtar, Tim M. Gale, Karen Irvine, Keith R. Laws.

**Validation:** Shazia Akhtar, Tim M. Gale, Karen Irvine, Dominique Grohmann, Keith R. Laws.

**Visualization:** Charlotte E. Dean.

**Writing – original draft:** Charlotte E. Dean.

**Writing – review & editing:** Charlotte E. Dean, Shazia Akhtar, Tim M. Gale, Karen Irvine, Keith R. Laws.

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
