## [Decision Letter · Decision Letter 0]

13 Dec 2021

PONE-D-21-32750Paranormal beliefs and cognitive function: A systematic review and assessment of study quality across four decades of researchPLOS ONE

Dear Dr. Dean,

Thank you for submitting your manuscript to PLOS ONE. After careful consideration, we feel that it has merit but does not fully meet PLOS ONE’s publication criteria as it currently stands.

The reviewers have made a very careful job, and provide quite a large number of recommendations regarding potential ways in which the manuscript could be improved. Most revisions seem doable, although they might require the paper to be substantially rewritten.

In view of the detailed reports (attached below), I will not reiterate all their points. Main suggestions, however, seem to regard to major themes. First, the reviewers demand a better justification of the categories used to classify studies. The reviewers (and I) understand that there will always be a certain degree of arbitrariness in classifying the studies identified, but the commonality between studies in the same category is not always obvious, and some studies seem to be classifiable in a different category.

And second, all reviewers have found the results quite difficult to follow, and not always sufficiently informative. Section-wise interim conclusions are probably necessary for the reader to get a clearer picture of the results in each area of research.

On the side of strengths, the reviewers have evaluated very positively some aspects of the methodology, including protocol preregistration, strict adherence to PRISMA guidelines, and the careful assessment of evidence quality in the reviewed studies. Please address, however, the reviewers' concerns regarding the decision to exclude studies with adolescents, and the total-score assessment of study quality

We look forward to receiving your revised manuscript.

Kind regards,

José C. Perales

Academic Editor

PLOS ONE

4. We note that this manuscript is a systematic review or meta-analysis; our author guidelines therefore require that you use PRISMA guidance to help improve reporting quality of this type of study. Please upload copies of the completed PRISMA checklist as Supporting Information with a file name “PRISMA checklist”.

Reviewers' comments:

Reviewer's Responses to Questions

**Comments to the Author**

1. Is the manuscript technically sound, and do the data support the conclusions?

Reviewer #1: Yes

Reviewer #2: Yes

Reviewer #3: Yes

Reviewer #4: Partly

2. Has the statistical analysis been performed appropriately and rigorously? 

Reviewer #1: N/A

Reviewer #2: N/A

Reviewer #3: N/A

Reviewer #4: N/A

3. Have the authors made all data underlying the findings in their manuscript fully available?

Reviewer #1: Yes

Reviewer #2: Yes

Reviewer #3: No

Reviewer #4: Yes

4. Is the manuscript presented in an intelligible fashion and written in standard English?

Reviewer #1: Yes

Reviewer #2: Yes

Reviewer #3: Yes

Reviewer #4: No

5. Review Comments to the Author

Reviewer #1: The authors present a review article on the association between paranormal beliefs and divergencies in cognitive function. The topic has been gaining relevance in the last decades in the research community and is of interest for a wide audience. Although I agree with the authors that a metanalysis would have been more valuable, I reckon that the heterogeneity in the studies published so far hinders that approach. In any case, I believe that a systematic review paper like the one presented will be of use for many researchers interested in this field during the following years.

Nevertheless, I have some concerns I believe merit clarification before recommending publication:

Line 87: I think there is a parenthesis missing (or one too many)

Line 95: I guess that this is a matter of personal preference (so you can ignore it), but I thought the sentences on line 95 to 100 (“While factors…”) were not very relevant for the study (which is already quite long and includes many references) so they could be deleted.

Line 101: While setting the case for the study of cognitive function and paranormal beliefs, the authors comment on studies of “other” kinds of beliefs, such as religious beliefs and conspiracist beliefs. Regarding the former, as far as I know, religious beliefs are, at least sometimes, considered one type of paranormal beliefs (see the Traditional Religious Beliefs dimension in the widely used questionnaire by Tobacyk). I think this should be clarified in the text. As for conspiracist beliefs, the authors could make use of the concept of “epistemically unwarranted beliefs” (see Lobato et al., 2014) to stablish links between these types of beliefs and maybe include some references to pseudoscientific beliefs too.

Line 114: Irwin’s review only included null findings?

Line 133: The aims of the larger study should be (at least briefly) explained.

Line 143: Were any unpublished works or theses included in the final list of studies. If so, this should be clarified through the description of the process and, at least, when discussing the final selection.

(around) Line 177: The “sought for retrieval” step appears in Figure 1, but I think it is not explained in the text.

Line 207: I do not think illusion of control and causal illusion should be categorized as perceptual biases. As far as I know, they are usually characterized as cognitive biases (see Matute, Yarritu and Vadillo, 2011), and they are very different to the other studies included in this section (visual noise studies).

Line 212: Here, I missed inclusion of Torres et al. (2020) (it appeared in a Scopus search using “paranormal belief” AND cogni*)

Lines 221 and 227: In which unit are these quantities expressed and how do they relate with the percentages in line 217

205: How were paranormal beliefs measured in each study?

Lines 287 and 295: Aren’t repetition avoidance tasks a measure of probabilistic reasoning? I think these studies would fit more comfortably in the next section.

Line 370: associations between paranormal beliefs and what?

Line 372: Here, I missed Barberia et al. (2018) (also appeared in a Scopus search).

Line 378: reference 125 is about an association between pseudoscientific beliefs and paranormal beliefs. I am not sure this should be included as a (cognitive) measure of critical thinking given the relation between the two types of beliefs. If the authors decide to keep it, then they should also consider other studies relating different types of unwarranted beliefs with each other (e.g., Lobato et al, 2014; Fasce and Pico, 2019…).

Line 439: In this section I missed Meyersburg et al. (2009) study, but then I couldn’t find it when I tried a search in Scopus so I guess it is ok.

Line 507: when discussing sampling representativeness, I think the sex/gender issue deserves some comment. Were samples composed of both men and women? were they balanced? At least in those studies recruiting Psychology students my guess is that more women would have been involved…

Line 528: I would appreciate a table or at least more information on the percentages of usage of each different test (not just Tobacyk and ad hoc questionnaires). It could be even worth providing information separated by each topic. Could differences regarding the association between critical thinking and paranormal beliefs be related to the use of different measures of belief endorsement?

Line 683: I think the authors should include in the discussion some comment on the implications of the fact that most of the studies analyzed are correlational.

Line 755: Regarding the proposal of the fluid-executive theory, the authors should describe ways in which this hypothesis could be tested.

Reviewer #2: The authors present a systematic review (without meta-analysis) of the literature relating paranormal beliefs with performance in different cognitive tasks, with a particular focus in the critical assessment of the quality of the studies conducted so far. Overall, I think that this can be a valuable resource for researchers working on this area and, hopefully, it will also contribute to improving the quality of future research. I do have some concerns, though, that the authors might want to address in the final version of the manuscript.

Perhaps my most important concern is that in the present version of the ms it is quite difficult to follow the results section. As the authors themselves acknowledge their classification of tasks is somewhat arbitrary because “such classifications are necessarily a simplification and are not intended to be a definitive organisation”. This is undeniable true, but even so some of the headings collate results from radically different tasks and at some points I couldn’t help thinking that some paragraphs actually belonged in a different section. Or, perhaps alternatively, the authors might want to provide some explanation for the logic behind including several tasks under a common heading. This is particularly problematic in the section now titled “Cognitive and perceptual biases”, which includes a wide range of phenomena from illusion of control (measured in learning tasks comprising hundreds of trials) to perception of faces and other stimuli. What’s the common feature underlying these different tasks? Is it the case that all (or most) of them are somehow related to the (illusory) perception of patterns where there is just noise? If that’s the case, I think the authors need to spell this logic out more clearly, possibly change the name of the section and perhaps move some of the tasks that do not fit well in this category to other sections. They might even want to consider dividing this section in two or more less heterogeneous sections.

I also found it slightly odd that the jump to conclusions task is included in this section, when it is actually more similar to some of the probabilistic reasoning tasks included in the following section.

Also, in the present version it is easy to get lost in the enumeration of Results, without getting a glimpse at the whole picture until the Discussion section. If the results section were briefer this would be ok, but given the length of the paper I think many readers would find it useful to be reminded occasionally of the interim conclusions that can be reached with the information provided in each subheading. In other words, I miss 1-2 concluding sentences providing an overview of the results found for each category of tasks.

The assessment of the quality of the studies plays a very important role in the manuscript and it is indeed a great contribution. But while reading it I had some concerns about the reduction of this information to a single “quality score” for each study. Although this is certainly common in meta-analytic literature, this approach has also been criticized, and rightly so in my opinion (e.g., Jüni, P., Witschi, A., Bloch, R., & Egger, M. (1999). The hazards of scoring the quality of clinical trials for meta-analysis. JAMA, 282, 1054–1060.) I would not ask the authors to change their approach, but I think it would be worth mentioning, even if briefly, that reducing the responses to these quality scales to a single score can be misleading and should be taken with caution.

In the introduction, the authors mention that the two most common tests to measure paranormal beliefs have good psychometric properties. But this is just one side of the story. These studies are trying to relate paranormal beliefs with performance in cognitive tasks that might not have such good psychometric properties. See, e.g., Hedge, C., Powell, G., & Sumner, P. (2018). The reliability paradox: Why robust cognitive tasks do not produce reliable individual dif- ferences. Behavior Research Methods, 50, 1166–1186. https://doi. org/10.3758/s13428-017-0935-1 This is important, because the observed correlation between any two measures is attenuated downwards if any of them (not just the paranormal belief scale) is unreliable. If the reliability of these tasks does not improve in future research (or sample sizes are not adjusted taking it into account) this will necessarily result in high heterogeneity (i.e., much more variance from one study to another) and small average effect sizes.

Regarding sample sizes, I was a bit puzzled to read that “overpower” can be a problem in this area. I am afraid I strongly disagree with this point of view. In my opinion, there cannot be such a thing as an excess of power. The authors justify this saying that “… large studies might also be over-powered and thus, detecting very small and possibly trivial effects”. This confounds hypothesis testing (whether an effect is significantly different from zero) with parameter estimation (what the exact size of an effect). The fact that many researchers take a significant result for a “relevant” result is the consequence of using statistical tests mindlessly and it would be an error to encourage researchers to use smaller-than-possible sample sizes as a solution. Regardless of whether you want to test the null hypothesis or know the exact size of an effect, a large sample size will always be helpful because it will reduce the uncertainty of your inferences. What researchers need to be reminded is that not everything that is statistically significant is important.

Minor comments

- If the goal is to represent a trend in time, I suspect that most readers would find Figure 5 easier to read if years are plotted in the x-axis.

- In the flow chart, I couldn’t understand how from 475 records screened you take 5 reports for retrieval, but then below you have again 84 reports originating from these 5. Something seems to be wrong or misleading in the flow of records.

Reviewer #3: This paper provides a comprehensive and thorough review of research into the relationship between paranormal beliefs and cognitive functioning. As the authors note, considerable time has passed since the last review of paranormal beliefs and cognition, so this paper makes a strong, important, and timely (if not long overdue) contribution to the literature. I also commend the authors for preregistering their PRISMA guidelines and for the level of detail and clarity they provide about how the systematic review was conducted. Overall, I think this paper will make a great contribution to the literature. However, there are some areas where I believe the paper could be improved which I have highlighted in detail below.

Major points:

1) Regarding the paper structure, I think it might be better to more clearly separate the study quality assessment results and discussion from the paranormal belief and cognition findings and theory results and discussion. The Introduction focuses on the key points, is clear and easy to follow, and provides an appropriate (broad) set up for the focus of the paper. Similarly, the Method section is clear, concise, and enjoyable to read. However, from the Results onwards the paper can be quite difficult to follow in places. For example, it goes from the Method to an outline of findings in the literature, then to sections on study quality assessment, then back to a discussion of cognitive deficits/differences, then onto open science (I think this section should be moved to be alongside the sampling issues and non-respondents), then back onto a summary of research findings and a proposal for a new theory etc.

This review paper is doing two things. Firstly, extracting and assessing the quality of the existing literature on paranormal beliefs and cognitive function. Secondly, it is also outlining and synthesising the findings from that literature (plus proposing a new theory/hypothesis for testing). These are two separate and quite distinct focuses. Therefore, I think it would be better to more clearly delineate them and instead present all of one aspect (quality assessment and relevant discussion) followed by everything relevant to the other (outline and synthesis of findings).

2) Related to the above point, the results generally provide a clear and comprehensive summary of the various findings that have been catalogued within each subsection. However, I think each subsection would benefit from an overall summary or synthesis that brings it all together. If you make the changes to the paper structure that I have recommended above, this may no longer be necessary because these results will be more closely followed by a relevant discussion section (but see how it looks and consider it). In comparison, the results for the quality assessment are accompanied by relevant discussion within the actual results section.

3) I think the proposed fluid-executive theory is underdeveloped and would benefit from some further explanation and elaboration. You explain how it would relate to probabilistic reasoning but don’t outline how it would contribute to or explain the other findings (e.g., cognitive and perceptual biases).

Additionally, I think you need to do more work to justify why this specific aspect should be focused on, rather than alternative explanations. For example, others might argue that analytical thinking (Pennycook et al., 2015), or a “rationality quotient” (Stanovich, 2016; Stanovich et al., 2016; Weller, 2017; although see Ritchie, 2017), could also be proposed as underlying (or overarching) theories that explain the various associations between paranormal belief and cognitive functioning. I am happy to be convinced that the proposed theory is the best/most plausible candidate, I just think it needs some further fleshing out and additional evidence to support it.

Minor points:

1) I understand the desire to not include studies on children, given the potential cognitive differences. However, from examining Table S1, it seems that for all the excluded studies the vast majority of participants included in the studies were adults and they just happened to also include some teenagers in the study. I think it would be justifiable to exclude studies that had solely focused on children or teenage samples but given the already wide variability in cognitive function between 18- and 70-year olds, it seems unnecessary to exclude these studies solely because they also include some participants in their mid-late teens. I know this is a deviation from your preregistration so feel free to push back or disagree, but it wouldn’t change your conclusions and I think it’s okay to make some deviations if they are well justified.

2) Is the discussion of test theories and differential item functioning in the Introduction necessary? It seems like an unnecessary distraction from the main focus of the paper, so I’d just leave it at a sentence or two explaining that there is debate about why these differences are found.

3) I think that Figures 2-4 for the AXIS data would be greatly improved if you also included No and Unsure (you could keep the current format but have the bars contain different colours for each response category). This is particularly important because when initially looking at the figures it is not clear that there was an “unsure” category (e.g., it looks like half the studies didn’t have ethical approval, when it likely just wasn’t explicitly reported).

4) The section on open science focuses solely on pre-registration but there are many other aspects of open science such as publicly sharing data, analysis scripts, materials etc. I think you should either broaden this section to cover those additional open science aspects or, if you want to avoid lengthening the paper, then you could combine the pre-registration section with the sample size justification section, presenting it as a possible solution to address these problems.

5) The data are not currently accessible via the OSF link (https://osf.io/7bthg/). Please update the OSF page to make it public or provide a reviewer only link if you don’t want to make the data fully open to the public yet. Don’t worry, I’ve done the same thing before and this happens with half or more of the OSF links I’ve seen when reviewing papers.

References mentioned in the review that are not already in the paper:

Pennycook, G., Fugelsang, J. A., & Koehler, D. J. (2015). Everyday consequences of analytic thinking. Current Directions in Psychological Science, 24(6), 425–432. https://doi.org/10.1177/0963721415604610

Ritchie, S. (2017). Review of: The rationality quotient: Toward a test of rational thinking (K. E. Stanovich, R. F. West, & M. E. Toplak). Intelligence, 61, 46. https://doi.org/10.1016/j.intell.2017.01.001

Stanovich, K. E. (2016). The comprehensive assessment of rational thinking. Educational Psychologist, 51(1), 23–34. https://doi.org/10.1080/00461520.2015.1125787

Stanovich, K. E., West, R. F., & Toplak, M. E. (2016). The Rationality Quotient: Toward a Test of Rational Thinking. The MIT Press. https://doi.org/10.7551/mitpress/10319.001.0001

Weller, J. (2017). Review of: The rationality quotient toward a test of rational thinking, by Keith E. Stanovich, Richard F. West, and Maggie E. Toplak. Thinking & Reasoning, 23(4), 497–502. https://doi.org/10.1080/13546783.2017.1346521

Reviewer #4: Overall Evaluation:

The paper presents a review and summary of the past 40 years of research on paranormal belief and cognitive functioning. As noted by the authors, there has not been a systematic review of this relationship since Irwin’s (1993) work almost 30 years ago. I wholeheartedly agree with the authors that such a systematic review is needed, and that it would add significantly to our overall understanding of the current state of the field. Unfortunately, though, I think there are some key issues with the current attempt that need to be addressed to turn it into a beneficial contribution to the area.

Comments:

1. Given the range of tasks and variables in this particular area, I have no doubt it was difficult to synthesize the information in a straightforward and simple manner. However, even though I work in this area, I found it hard to track through the main sections. Specifically, each section was a listing of how one study showed X, two studies showed Y, et cetera, and by the end of each section it was not clear what specifically the reader should take away. At a minimum, using something like clear and specific tables to help organize the material would help immensely, especially in terms of trying to track through what the various studies do or do not show. There are the tables in the supplementary material, and admittedly even though they are broken up by section and a bit tricky to see “overall” outcomes, I found them easier to follow in terms of thinking across the studies.

Relatedly, in several spots the writing/presentation was dense, which may have added to the experience of not knowing what the “take home” message was for each section. For example, proper paragraphs should rarely run over a page, but more than one did, and one paragraph actually went for almost a full 2 pages (pgs 11-13). In general, editing for direct language, paragraph length, et cetera, would help improve clarity of the information being presented.

2. Again, I understand it would be difficult to categorize the experiments, but the current way of doing it seems to actually work against providing a systematic review. For example, there were fewer studies than I would have expected in the thinking styles section, just because this has been a particularly popular topic to explore in terms of paranormal beliefs. I could see, though, how some of that work would have ended up in other sections given the classification criteria; however, it then feels like we are not getting the full picture. Again, I think this is where tables may be particularly useful; for example, rather than binning studies under just one section, it would be much more useful to have tables that include all of the categories. Thus, we would be able to see what each study contributes across the categories (when relevant), rather than to just a single category. I recognize the authors may have attempted this approach and for some reason it was not viable, but based on systematic reviews in other areas that have used this set-up it would seem to be the more useful approach. This type of set-up would also help more clearly and succinctly demonstrate what the reader should take away from the area.

3. The limitations discussion seems like a bit of a tacked-on section rather than a real consideration of the current work. For example, as already mentioned, one potential issue is how the studies had to be categorized, which means they can only contribute to one section even though they may potentially also be able to contribute to at least one other section.

Further, and this may be an unfair criticism given the complexity of the area, but I had fully expected at least some sort of meta-analysis of the studies. Again, this may be because the listing out of the studies across the sections did not land well in terms of what to concretely take away. However, given the current techniques available for meta-analyses it feels like that is what we would gain the most from in terms of understanding the current state of the relationships between paranormal belief and cognitive functioning.

4. There seems to be some sweeping generalizations made that are not necessarily an accurate representation of the area. For example, there’s a difference between a cognitive deficits hypothesis suggesting “paranormal believers are illogical, irrational, and uncritical,” and what the researchers of each study argued for as the hypothesis/explanation for their work. Sure, some likely would subscribe to this particularly spin on cognitive deficits, but looking for “more or less” of a skill/ability does not necessarily mean researchers in this area would agree that believers are “illogical” or “irrational”.

5. A minor point, but was any effort put into searching for articles using the term “anomalistic”? The term “paranormal” is still the most common terminology, but given Chris French and colleague’s focus on using the more broad term of anomalistic belief (and subsequent work from others that has followed suit), it is not clear whether just using “paranormal” would have picked up all of the relevant studies.

6. PLOS authors have the option to publish the peer review history of their article (what does this mean?). If published, this will include your full peer review and any attached files.

Reviewer #1: No

Reviewer #2: No

Reviewer #3: **Yes: **Toby Prike

Reviewer #4: No

---

## [Author Response · Author response to Decision Letter 0]

31 Jan 2022

We would like to thank the reviewers for their careful review and the insightful and detailed comments they provided. 

We have addressed and responded to the reviewers' comments, details of which can be found in the 12-page Response to Reviewers document.

We have addressed the main points raised, particularly the clarity of the results and the classification of studies, by providing new sections within the manuscript (e.g., summaries following each subsection of the results) as well as new supplementary materials (e.g., S9 Table).

We would like to thank the editor and reviewers again for the valuable comments, which we feel have greatly improved the manuscript, and for the opportunity to revise and resubmit the work.

---

## [Decision Letter · Decision Letter 1]

8 Mar 2022

PONE-D-21-32750R1Paranormal beliefs and cognitive function: A systematic review and assessment of study quality across four decades of researchPLOS ONE

Dear Dr. Dean,

Thank you for submitting your manuscript to PLOS ONE. The revised paper has been assessed by the same four reviewers from the previous round. All of them recommend minor revisions, but they are not fully coincident, so some amount of work is still required. Still, all suggested changes are modest and doable, so a further review round with the four reviewers again might not be necessary if all concerns are addressed.

We look forward to receiving your revised manuscript.

Kind regards,

José C. Perales

Academic Editor

PLOS ONE

Journal Requirements:

Reviewers' comments:

Reviewer's Responses to Questions

**Comments to the Author**

1. If the authors have adequately addressed your comments raised in a previous round of review and you feel that this manuscript is now acceptable for publication, you may indicate that here to bypass the “Comments to the Author” section, enter your conflict of interest statement in the “Confidential to Editor” section, and submit your "Accept" recommendation.

Reviewer #1: (No Response)

Reviewer #2: All comments have been addressed

Reviewer #3: (No Response)

Reviewer #4: (No Response)

2. Is the manuscript technically sound, and do the data support the conclusions?

Reviewer #1: Yes

Reviewer #2: Yes

Reviewer #3: Yes

Reviewer #4: Yes

3. Has the statistical analysis been performed appropriately and rigorously? 

Reviewer #1: N/A

Reviewer #2: N/A

Reviewer #3: Yes

Reviewer #4: N/A

4. Have the authors made all data underlying the findings in their manuscript fully available?

Reviewer #1: Yes

Reviewer #2: Yes

Reviewer #3: Yes

Reviewer #4: Yes

5. Is the manuscript presented in an intelligible fashion and written in standard English?

Reviewer #1: Yes

Reviewer #2: Yes

Reviewer #3: Yes

Reviewer #4: No

6. Review Comments to the Author

Reviewer #1: I am generally satisfied with the responses offered by the authors to my previous comments. Now I have some minor concerns, mostly regarding the PRISMA protocols, which I describe in the following:

PRISMA for abstracts:

- The Methods section should specify the inclusion and exclusion criteria for the review, and the dates when each database was last searched. It should also specify the methods used to assess risk of bias in the included studies.

- The Results section should indicate the number of included studies and participants for each relevant outcome mentioned (e.g., association between paranormal belief and intuitive thinking bias).

- The primary source of funding for the review should be specified.

PRISMA checklist (manuscript)

- Item 5: the Methods section should specify how studies were grouped for the synthesis

- Item 10: the Methods section should specify whether all results that were compatible with each outcome domain in each study were sought (e.g. for all measures, time points, analyses), and if not, the methods used to decide which results to collect.

- Item 11/12/14: If I understand it correctly, description of methods for assessment of risk of bias should be presented in the Methods section. Now they are described in the Results section.

Item 13/15: I could not identify descriptions corresponding to these items in the Methods section.

- Item 27: Report which of the following are publicly available and where they can be found: template data collection forms; data extracted from included studies; data used for all analyses; analytic code; any other materials used in the review.

-----

I like the proposal of the fluid-executive model, but isn’t the general idea that “it is possible to view the association between the many cognitive deficit-paranormal belief associations as the product of a single underlying fluid intelligence-executive component” in conflict with the fact that findings presented in the intelligence section “are highly conflicting, with an almost an equal number of significant versus non-significant findings”?

-----

I think S1 Appendix is not referred in the manuscript

Line 97: parenthesis missing

Line 420: I think the role of the beads task in Prike et al.’s study in unclear. In addition, this task is not explained until the next paragraph. Maybe changing the order of those two paragraphs (or explaining the task the first time it is mentioned) would help the reader to understand its relevance.

Line 549: revise “statements”

Line 568: revise “be”

Line 692: I think “such” is not appropriate there given we have just started a new section

Reviewer #2: The authors have done an excellent job at addressing my concerns with the previous version of the manuscript. I only have a few minor comments:

- Line 5, there is a parenthesis missing at the end of the line.

- Lines 215-216. Please say explicitly what you mean by large, moderate and small effects.

- Page 12, first paragraph: any study published as a registered report? This would be interesting because RRs do not only limit p-hacking, they also ensure that that there is no publication bias (papers are accepted or rejected before results are known).

- Lines 412-414. “Paranormal believers showed a lower perceptual sensitivity compared to sceptics (a bias towards making more ‘yes responses…” Sensitivity and bias are completely different things (e.g., in signal detection theory analysis). Please, clarify whether believers differ in one or the other.

- Line 556 “… conducted by similar research teams”. Similar in what sense?

Reviewer #3: The authors have been very receptive to the comments made in the previous round of reviews and the revised manuscript is much improved. I would like to again highlight that this review of paranormal beliefs and cognition makes a strong and timely contribution to the literature. I have highlighted a few minor points below which I believe can easily be addressed.

Minor points:

1) The sentence on page 13, lines 263-266, discusses differences between studies with student and non-student samples but the analyses reported do not find significant differences. Please make the lack of significant differences clear and adjust/remove the related discussion.

2) When discussing the conjunction fallacy (pages 22 and 25), in addition to Rogers et al., Prike et al. (2017) also found a significant relationship between the conjunction fallacy and paranormal belief. Additionally, the differences between the studies may be due to differences in analysis techniques used. In the Dagnall et al. studies, all the probabilistic reasoning tasks were entered together as predictors, whereas Rogers et al. and Prike et al. looked at the relationship between the conjunction fallacy and paranormal belief without entering/controlling for these other probabilistic reasoning tasks, which may explain the differences in results. This doesn’t need much discussion but may be worth noting or mentioning.

3) The section on page 31, lines 701-705, is unnecessarily repetitive and makes the same point multiple times (generally study quality is good but there are some specific areas for improvement).

4) On page 32, line 731, the text says “Eight” but the parentheses say “(9/71)”.

Reviewer #4: Overall Evaluation:

The authors did a good job addressing the issues raised by all of the reviewers, and overall I do think those revisions make for a much clearer and easier-to-follow narrative, and thus a stronger manuscript. I also appreciated the additional data/info included, such as giving a clear overview of the alternate categories in S9. I do still have a few comments, but given the focus of the paper I do not think any of them are major issues, and I also understand their reasoning for some of the issues they chose not to address with changes in the manuscript (e.g., as much as I would love to see some meta-analyses stats, I do understand the authors’ reasons for choosing not to go that route).

Comments:

1. On pg 13 (lines 263-266) claims are made that aren’t supported by the provided statistics. So either those statistics are incorrect (or I am misunderstanding what is being reported), or the wording needs to be changed. That is it cannot be said that the undergrad studies tended to have smaller samples and lower quality; however, it could be said that descriptively there looks to be a difference but there is no statistical evidence that there is indeed a difference.

2. I appreciate the expanded limitations section, but still think there are issues with it. First, some of the points are underdeveloped; for example, the discussion about weighting vs. summing on the AXIS needs to be unpacked with even just 1-2 more sentences to be clear what specifically the issue may be (pg 120, lines 863-867). Admittedly I am a bit old-school and still believe in the rule that proper paragraphs are a minimum of 3 sentences, but here I am saying it because the 2 sentences that are there do not follow-through the point, especially for readers who may not be familiar with the specific issue.

Further, I expected the issue of categorization to be included in the limitations, so was quite surprised when I saw it was not there. It is mentioned earlier in the manuscript, which was good, but the alternate categories table (S9) shows that it warrants further discussion in terms of limitations. That is, the table very nicely shows just how many potential categories the majority of studies could fall into. So I want to be clear that I am not trying to argue that the review be redone with the studies being included in all possible categories because I understand the author’s justification for the way they decided to do the categorization. However, I do think more consideration is needed for how this effects what we can conclude for each section (i.e., given how many “secondary focus” studies had to be excluded from consideration because they were used elsewhere).

3. I think the summary sections are quite useful, and go a long way to helping to track through what each section is trying to convey. However, that (along with other additions) does mean that overall the manuscript is a bit daunting to get through, and I think in general it feels like it has a lot of repetition in some spots. For example, the start of the General Discussion is mainly just repetition (including all the numbers again) of what has already been presented. A lot of that could be deleted or streamlined into main points that move beyond what has already been said in earlier sections. And I would also argue focusing on my direct and active writing would cut out a lot of wordiness and also help the manuscript feel more streamlined and manageable (including cutting down on the large amount of information given in parentheses).

4. There is an issue with the wording of the sentence on pg 88, lines 99-102. The first half is hard to follow due to the two “was” near each other, and I think at least one of those needs to be reworked to help clarify what is being said.

7. PLOS authors have the option to publish the peer review history of their article (what does this mean?). If published, this will include your full peer review and any attached files.

Reviewer #1: No

Reviewer #2: No

Reviewer #3: **Yes: **Toby Prike

Reviewer #4: No

---

## [Author Response · Author response to Decision Letter 1]

4 Apr 2022

We would like to thank both the editor and reviewers for their detailed comments, and for the opportunity to submit a re-revised version of the manuscript. A point-by-point response to the reviewers' comments can be found in the attached 'Response to Reviewers (2)' file. We have made edits throughout the manuscript to improve both the length and clarity of the manuscript and the reviewers' advice, which we feel have greatly benefitted the manuscript. We would like to thank the editor and reviewers again for the time they have committed to reviewing this manuscript.

---

## [Editor Report · Decision Letter 2]

7 Apr 2022

Paranormal beliefs and cognitive function: A systematic review and assessment of study quality across four decades of research

PONE-D-21-32750R2

Dear Dr. Dean,

We’re pleased to inform you that your manuscript has been judged scientifically suitable for publication and will be formally accepted for publication once it meets all outstanding technical requirements.

Kind regards,

José C. Perales

Academic Editor

PLOS ONE
---

## [Editor Report · Acceptance letter]

12 Apr 2022

PONE-D-21-32750R2 

Paranormal beliefs and cognitive function: A systematic review and assessment of study quality across four decades of research 

Dear Dr. Dean:

I'm pleased to inform you that your manuscript has been deemed suitable for publication in PLOS ONE. Congratulations! Your manuscript is now with our production department. 

Kind regards, 

on behalf of

Dr. José C. Perales 

Academic Editor

PLOS ONE